# Visualization of Activated Area on Polymers for Evaluation of Atmospheric Pressure Plasma Jets

**DOI:** 10.3390/polym13162711

**Published:** 2021-08-13

**Authors:** Dariusz Korzec, Thomas Andres, Eva Brandes, Stefan Nettesheim

**Affiliations:** Relyon Plasma GmbH, Osterhofener Straße 6, 93055 Regensburg, Germany; t.andres@relyon-plasma.com (T.A.); e.brandes@relyon-plasma.com (E.B.); s.nettesheim@relyon-plasma.com (S.N.)

**Keywords:** atmospheric pressure plasma jet, dielectric barrier discharge, piezoelectric direct discharge, surface free energy, test ink, surface activation

## Abstract

The treatment of a polymer surface using an atmospheric pressure plasma jet (APPJ) causes a local increase of the surface free energy (SFE). The plasma-treated zone can be visualized with the use of a test ink and quantitatively evaluated. However, the inked area is shrinking with time. The shrinkage characteristics are collected using activation image recording (AIR). The recording is conducted by a digital camera. The physical mechanisms of activation area shrinkage are discussed. The error sources are analyzed and methods of error reduction are proposed. The standard deviation of the activation area is less than 3%. Three polymers, acrylonitrile butadiene styrene (ABS), high-density polyethylene (HDPE), and polyoxymethylene (POM), are examined as a test substrate material. Due to a wide variation range of SFE and a small hydrophobic recovery, HDPE is chosen. Since the chemical mixtures tend to temporal changes of the stoichiometry, the pure formamide test ink with 58 mN/m is selected. The method is tested for the characterization of five different types of discharge: (i) pulsed arc APPJ with the power of about 700 W; (ii) piezoelectric direct discharge APPJ; (iii) piezoelectric driven needle corona in ambient air; (iv) piezoelectric driven plasma needle in argon; and (v) piezoelectric driven dielectric barrier discharge (DBD). For piezoelectrically driven discharges, the power was either 4.5 W or 8 W. It is shown how the AIR method can be used to solve different engineering problems.

## 1. Introduction

The non-equilibrium atmospheric pressure plasma (APP) is broadly used for polymer surface treatment [1,2,3], especially for the improvement of adhesion [4]. Even though it is a long-known technology, the surface treatment of polymers remains the focus of current research. The application examples are the improvement of adhesion on composites of polymers with natural materials [5] or applications in medical sciences, for example, tailoring of the cell growth [6].

Due to its excellent chemical and mechanical properties, polyethylene (PE) is a widely used engineering material. Its drawback is its low surface free energy (SFE) of 31–37 mJ/m${}^{2}$ [7,8]. The APP is frequently used to increase the SFE of PE, to allow printing, varnishing, coating, or gluing on its surface. Examples of studies focussing on this subject include the application of dielectric barrier discharge (DBD) for the treatment of high-density polyethylene (HDPE) [9] or ultrahigh-modulus polyethylene (UHMPE) fibers [10].

A very versatile method of APP generation involves using atmospheric pressure plasma jets (APPJ) [11,12]. There exist a large number of gaseous discharge architectures used for such processing tools [13]. The APPJs are also used to increase the SFE of polymers [14,15] and, consequently, to improve the adhesion of glues and molds [5], or printability [16]. There is increasing interest in biomedical applications of cold APPJs [17]. The mechanisms of the surface modification by atmospheric pressure plasma jets (APPJ) are still not completely understood [18]. Different types of APPJ are used for the treatment of PE, for example, the high voltage blown arc [14,19], the capacitive, cold APPJ [20], DBD driven APPJ [21], or piezoelectric direct discharge (PDD) [22]. Several gases are used for APPJ treatment of PE, for example, Argon [23,24] or compressed dried air (CDA) [19]. APPJ improves adhesion to PE in 3D bio-printed structures [25] or modifies the surface properties of shoulder implants made of ultra-high molecular weight polyethylene (UHMWPE) [26]. Compared with traditional polymer surfaces activation methods, such as flaming [27,28] or corona treatment [29,30,31,32,33], the APPJs allow the achievement of locally much higher values of surface free energy.

The diversity of APPJ physics and chemistry makes it difficult to evaluate and compare the efficiency of different APPJs using conventional plasma plume diagnostic techniques, such as electrostatic probe diagnostics [34], dielectric probes [35], optical emission spectroscopy [36,37], absorption spectroscopy between VUV to MIR [38], electron density measurements by millimeter wave interferometry or by IR heterodyne interferometry [39], high-speed photography [40,41], or calorimetric probe [42].

A pragmatic approach for comparative plasma source evaluation is to establish a model process and define a measurable process parameter, for example, deposition rate, etch rate, or activation speed. In this work, the activation area reached after some predefined treatment time on the polymer surface is proposed as such a comparative parameter.

To determine the activation area, a novel activation image recording (AIR) technique is developed. The test inks are investigated in a wide range of the SFE and the optimum ink is selected. Three polymers, acrylonitrile butadiene styrene (ABS), high-density polyethylene (HDPE), and polyoxymethylene (POM) are examined as materials for test substrates. The tests with HDPE show its suitability for AIR.

This study aims to demonstrate that AIR is suitable for the comparison of APPJs belonging to different power classes. On the one hand, the low power (8 W) room temperature PDD type of plasma is used [22]. On the other hand, the powerful (700 W) pulsed atmospheric arc (PAA) type of APPJ is applied [43]. The suitability for different types of APPJ discharges and different gas mixtures should also be evaluated.

## 2. Materials and Methods

### 2.1. Atmospheric Pressure Plasma Sources

In this study, the activation area was produced using atmospheric pressure plasma (APP) sources with seven discharge configurations, marked with A to G, illustrated in Figure 1. The typical operation parameters for all configurations are summarized in Table 1.

In configuration A, the pulsed atmospheric arc (PAA) was used for plasma generation. Its operation principle is explained in detail in [43]. As an example, the plasmabrush^®^ PB3 of Relyon Plasma GmbH with A450 nozzle was used. This APPJ is an industrial device designed for the high-speed treatment and allows control of the plasma ON-time in the ms range.

The configurations B, C, and D are all based on the piezoelectric direct discharge (PDD), described in detail in [22], but representing different technical solutions. The configurations B and C were used in the commercial hand-held devices piezobrush^®^ PZ2 and piezobrush^®^ PZ3 of Relyon Plasma GmbH, respectively. Both applied the piezoelectric transformer CeraPlas™F [44] for high voltage generation. The main differences between B and C in the physical sense are the different air flows (see Table 1). For some experiments, the PDD was generated using the CeraPlas™HF (discharge configuration D) [45], which is smaller than CeraPlas™F. Consequently, its maximum operation power was lower and the operation frequency was higher (compare the operation parameters in Table 1).

In configuration E, a needle corona [46] was supplied with high voltage (HV) from the CeraPlas™F over a plasma bridge [47]. This HV causes an atmospheric air corona discharge on the tip of the needle electrode.

The configuration F, referred to as a plasma needle [48], consisted of the same needle electrode as in configuration E, but was operated in a gas flow of different gas mixtures [47]. In this study, the results obtained with argon as a plasma gas are presented.

The discharge configuration G produced a dielectric barrier discharge (DBD) with an active electrode being the HV tip of the CeraPlas™F. Configurations E, F and G are realized technically as replaceable modules for piezobrush^®^ PZ3 [47].

Figure 2a shows a generic setup for substrate surface activation, with the distance *d* between the treated surface and the plasma source. The meaning of this distance differs depending on the configuration. The plasma source reference position for configuration A is the tip of the copper nozzle A450 of the plasmabrush^®^ PB3. For configurations B, C, and D, this reference is the tip of the piezoelectric transformer CeraPlas™, for configurations E and F, the tip of the needle powered piezoelectrically, and for configuration G, the outer surface of the dielectric barrier.

### 2.2. Visualization of the Activation Area

The activated zone on the surface of the polymer substrate produced using one of the discharge configurations listed in Table 1 can be visualized by spreading a test ink over the substrate surface. The properly chosen value of the test ink (see Section 2.4.3) assures that the ink is wetting only the activated surfaces and rolls off the non-treated surface areas.

The plasma source was positioned at a given distance from the substrate surface and the power was switched on for a short predefined time. Figure 3 shows the shapes of the visualized activation area produced using discharge configuration A, C and G. As can be expected, the rotationally symmetric configuration A produces a rotationally symmetric activation zone. The difference between pictures A${}_{1}$ and A${}_{2}$ is in treatment time, which is 100 ms and 3 s, respectively. In the case of A${}_{2}$, the treatment time is so long, that the thermal damage in the center of the activation zone occurs, resulting in the reduction of the SFE.

The pictures C${}_{1}$ and C${}_{2}$ show the activation zones produced in configuration C. The activation zone visualized in picture C${}_{1}$ was produced by a 10 s treatment at a distance of 4 mm. The kidney-like shape was caused by the shape of the discharge itself, following the rectangular geometry of the CeraPlas™F HV tip. The image in C${}_{2}$ was obtained at the much larger distance of 10 mm and the shorter treatment time of 1 s, resulting in the splitting of the activation zone into two small sub-zones. The activation images produced using configurations B and D are very similar to the image for configuration C because they were also generated by the piezoelectric device of CeraPlas™ type.

The pictures G${}_{1}$, G${}_{2}$ and G${}_{3}$ show the activation areas produced in configuration G. The characteristic four-folded shape in G${}_{2}$ is caused by the four fixtures used to keep the quartz dielectric barrier cup. The pictures G${}_{1}$ and G${}_{3}$ demonstrate how the tilting of the plasma handheld instrument affects the shape of the activation area. The tilting was performed in the direction perpendicular to the narrow and to the wide sides of the CeraPlas™F, respectively.

The circular activation areas related to the rotationally symmetric needle electrodes of configuration E and F are not shown in Figure 3.

### 2.3. Activation Image Recording

The shape of the activation area provides interesting information about the character of the plasma source used for the activation. However, for quantitative evaluation of the plasma source performance, some reference value must be defined. In this study, the area of the activation visualized by the test ink is such a value. To increase the accuracy and repeatability of the evaluation of the test ink patches, the automated method of ink patch area determination was introduced. Since the area of the ink patches changes over time (see Section 3.1.1) the pictures of the ink patch were taken in short intervals with a digital camera (see Figure 2b). The contour of the ink patch was automatically recognized and the number of pixels was counted. By comparison with the number of pixels of the known area of the entire substrate, the actual area of the test ink patch was calculated. The algorithm applied in the activation image recording (AIR) the method has some limitations. The image recognition was optimized for dark blue ink on a white substrate. If the test ink was of another color or was too pale, errors in recognition of the ink patch boundary could occur. Such errors are also possible if the substrate was dark or if the illumination of the ink patch was weak or not time-stable. In the case of splitting of the activation area into more than one patch, as shown in C${}_{2}$ in Figure 3, the largest sub-patch was selected for evaluation. The not inked areas within the ink patch, such as the inner circle shown in A${}_{2}$ in Figure 3, were not subtracted.

### 2.4. Test Inks

#### 2.4.1. Formamide Based Test Inks

Different liquid mixtures were used for the production of test inks suitable for the determination of the SFE of solid surfaces. The test inks defined in a number of standards [49] and gauging the surface energies from 31 to 58 mN/m were mixed with formamide and 2-ethoxyethanol (alternative names: ethylene glycol monoethyl ether or ethyl Cellosolve—registered trademark of Union Carbide Corp.). The dependence of the gauged SFE on the volume percentage of formamide in the test ink mixture is shown in Figure 4a. For gauging the test inks in the SFE in the range from 58 to 72 mN/m, the mixtures of formamide with DI water were used. The linear approximation of the SFE on the volume percentage of fomamide in a formamid–water mixture is also shown in Figure 4a.

#### 2.4.2. SFE Radial Distribution

Since the chemically active species causing the surface activation were not distributed homogeneously across the plasma plume of an APPJ, the SFE also varies across the activated zone. Consequently, applying the test inks gauged with different SFEs, different sizes of the activation zone were obtained. Figure 4b shows the radial distribution of the SFE on an intensively treated HDPE. The size of the visualized activation zone almost doubled if the test ink 46 mN/m was taken instead of the pure water-based test ink 72 mN/m. On the one hand, the relative accuracy of the AIR method improves with the size of the activation zone. On the other hand, a too low SFE of the test ink results in large visualized areas even for poor activation. The SFE value of 58 mN/m is a good compromise between these two extremes. The intensive treatment means that a large part of the activation area reaches the saturation value of 72 mN/m. For weak treatment, the SFE distribution curve could be below the 58 mN/m resulting in no visualized activation area. The SFE distributions reaching just over the 58 mN/m line would be not suitable, because the visualized area would be very sensitive to any random influences and the results would be unreproducible. The assumption for the validity of the AIR method is that the treatment time is long enough to assure the intensive treatment.

#### 2.4.3. Aging of Test Ink

Since the 2-ethoxyethanol is more volatile than the formamide, the stoichiometry, and consequently the SFE gauged by test ink mixed from formamide and 2-ethoxyethanol, can increase over time. It corresponds to the movement to the right along the blue line in Figure 4. This test ink will show a smaller activation area than the not aged test ink. No such change of composition is exhibited by the 58 mN/m test ink, because it is made of 100% formamide. The pure formamide has a surface tension of 58.2 mN/m but the commercially available inks are specified with some tolerance, typically ±0.5 mN/m. No sensitivity to stoichiometry was an important reason for the selection of this specific ink as a standard for this study.

Despite using the test ink consisting of a single liquid chemical, strong differences in the visualization result can be observed when older formamide test inks are used. The activated areas visualized with test ink from vials differing in age by more than 2 years differ by more than 25%. On the other hand, the results obtained with fresh ink from different vials are very reproducible with the relative standard deviation of visualized areas of less than 3%.

The other origin of the instability of formamide based test inks is their hygroscopicity [50]. When absorbing moisture from humid air, the test ink dyne number shifts in the direction of higher values causing a decrease of the visualized surface. It corresponds to the movement to the left along the red line in Figure 4.

To avoid the aging of the ink, only fresh ink vials were used and the vial remained open only for a short time of ink application.

#### 2.4.4. Environmental Influences

It is known that environmental conditions, such as air temperature, relative humidity and pressure, have a significant influence on the wetting properties of liquids on polymer surfaces. The liquid temperature has a significant influence on its surface tension. The temperature coefficient of surface tension is −0.1514 and −0.0842 mN/(K·m) for water and formamide, respectively. Consequently, a variation of temperature of more than 10 K would cause the change of the calibration number of the formamide test ink.

It is documented in the literature that the increase of the pressure causes an increase of the contact angle [51], which manifests in a decrease in the activation area.

The humidity influences both the surface tension of the liquid water–air interface [52] and the water contact angle on the solid surface [53].

To evaluate the influence of the environmental factors on the activation area results, these conditions are logged in during the test ink patch evaluation. The temperature, humidity, and pressure sensors are placed at a distance of 10 cm from the optical axis of the digital camera (see Figure 2b).

### 2.5. Substrates

The substrates used in this study are made of three pristine polymers: acrylonitrile butadiene styrene (ABS), high-density polyethylene (HDPE), and polyoxymethylene (POM) and are delivered by Rocholl GmbH, Germany. It is known that additives, such as stabilizers, have a strong influence on the surface properties of polymers [54]. For test substrates, the “natural” HDPE without additives were used. For plasma treatment and activation area evaluation the substrates with sizes 100 mm × 50 mm × 2 mm or 50 mm × 50 mm × 2 mm were used. The second one was assumed if no explicit information was included. For plasma treatment, the substrates were fixed on a solid block of HDPE with sizes 100 mm × 100 mm × 10 mm. Only the substrates treated in discharge configuration G were placed on a grounded metal plate.

According to the standard [49], the substrates were preconditioned after delivery at least 40 h under 23 °C and 50% humidity. No cleaning procedure was applied. All substrates were treated with plasma and exposed to the test ink at temperature 23 °C ± 2 K and a relative humidity of 50% ± 5%.

## 3. Results and Discussion

The results consist of two parts. In the first part, different influences on the visualized activation area are analyzed and the definition of the activation area, as used for the evaluation plasma sources’ performance, is proposed. In the second, the proposed definition of the activation area is implemented to evaluate different types of APP sources.

### 3.1. Activation Area Determination

#### 3.1.1. Shrinkage of the Test Ink Patch

As Section 2.3 stated, the activated area wetted with test ink decreases significantly with time. The strongest change is observed within a few seconds after application of the test ink. The stable value is reached within minutes. The dependence of the visualized activated area on time elapsing after an ink application, as shown in Figure 5a, will be called shrinkage characteristics.

In this example, the shrinkage speed of the test ink patch decreases with time. In the first second, the ink patch area decreases by 8%. At 7 s after wetting, the shrinkage speed is about 1.4% per second. At 2 min after wetting, it is only 0.11%/s. As a reference time for the AIR method, the time of 10 s with shrinkage speed of 0.8%/s is set. It is a compromise between a large error in the surface determination for very short times and the influence of material and environmental factors for very long times.

The activation area visualized after 10 s of shrinking corresponds, not exactly, to 58 mN/m. Due to the water absorption in the ink after 10 s, the ink shows the wetting of surfaces with higher surface free energy than 58 mN/m. The value for t=0 s would correspond to exactly 58 mN/m because the test ink has this property immediately after application only. The application recipes for the commercial test inks prescribe that the evaluation of the ink wetting should be conducted within 2 s after application, accepting the disadvantage of the very fast-changing of the ink wetting properties in this time range.

#### 3.1.2. Selection of Substrate Material

The selection of the substrate material has a strong influence on the accuracy and activation area range of the AIR method. On the one hand, it is important that the selected material should have the SFE without plasma treatment far below the value of 58 mN/m. On the other hand, the maximum SFE reachable after the plasma treatment should be significantly higher than 58 mN/m.

The formamide based test inks are “blind” on the polar component of activation on some plastics; an example is a popular polymer PVC (polyvinyl chloride) [55]. Such materials can be excluded as a standard for plasma sources’ evaluation. Another group of materials, not suitable for this purpose, is that exhibiting a strong hydrophobic recovery [56,57]. The materials that significantly lose their hydrophilicity obtained by plasma treatment after minutes are, for example, thermoplastic polyurethane (TPU) [58] or polydimethylsiloxane (PDMS) elastomer [59]. A comparatively weak hydrophobic recovery exhibits polyethylene (PE), especially if treated with oxygen-containing plasmas [60]. Its native contact angle with deionized water is about 90° and the SFE of the HDPE substrates used in this study is 36 mN/m. The measurements conducted in our lab with the use of a Krüss droplet test instrument show that the SFE of 66 mN/m and the contact angle of 34° reached after brief plasma treatment remains unchanged after 4 h in ambient air. The contact angle increases in 100 days by 7°, which is much less than for many common polymers. For a change of activation area with storage time after treatment—see Section 3.1.6. Our measurements on ABS substrates plasma-treated in configuration A also show no measurable hydrophobic recovery 4 h after the plasma treatment.

Due to the wide range of SFE after plasma activation and low hydrophobic recovery, the ABS, POM and HDPE are suitable substrate materials for the AIR method. To select the best material among these three, the shrinkage characteristics for these materials were collected. The results are shown in Figure 5b. Two disadvantages of POM can be stated. First, the activation area of POM is much lower than that of ABS and HDPE, causing the reduced resolution of the AIR technique. The second is the speed of variation of the activation area. The ratio of starting area to the area after 10 s of shrinking is 2.07 for POM, compared with 1.35 and 1.25 for ABS and HDPE, respectively.

The possible explanation of the strong variation of the shrinkage characteristics for POM can be its strong water absorption reaching up to 0.5% of water in comparison with HDPE, absorbing only up to 0.01% of water. In the case of POM, the water from the substrate can be absorbed by the hygroscopic formamide, increasing the gauged surface free energy and consequently decreasing the visualized activation area.

The high sensitivity on plasma activation makes ABS advantageous as a reference substrate for the evaluation of weak plasma sources. The well-defined pure water test ink with 72 mN/m can be used in combination with ABS substrate. The red curve in Figure 6a shows the activation area on the ABS surface generated in configuration G as a function of the SFE value of the test ink. The curve shows that, after only a 10 s treatment, a very small variation of the activation area of 7% is observed when changing the SFE of the activation area definition from 46 to 72 mN/m. The consequence is the reduced range of activation area available for the AIR method. A small difference between strongly and weakly activated areas can be expected. In contrast to ABS, the HDPE substrates show a very strong variation of the activation area with SFE value of the test ink, more than doubling, when the SFE value of the test ink decreases from 72 to 46 mN/m. A similar difference is observed for HDPE activated in configuration C (see Figure 6b). This outstanding range of the activation area on HDPE was an important reason for the selection of HDPE as a standard material for the AIR method.

#### 3.1.3. Ink Patch Shrinkage Mechanism

In this section, results are presented that support the thesis that the increase in water concentration in the formamide test ink layer is the main reason for the ink patch shrinkage.

During the collection of shrinkage characteristics, the relative humidity in the substrate vicinity was raised intentionally by about 10% for 5 s. During this time, a strong increase in the shrinkage speed can be observed (slope line 2 in Figure 7a). This result confirms the shrinkage mechanism based on the increase of water content in the ink. A significantly smaller shrinkage speed after exposure to higher humidity (slope line 3 in Figure 7a) than before exposure (slope line 1) can be explained by this effect. Since during the exposure of the ink to air with higher humidity, a higher concentration of water in ink is reached, after establishing the previous value of humidity, the shrinkage speed is lower because the concentration gradient of water in the air and the ink is lower.

The amount of applied test ink has a significant influence on the shrinkage characteristics. In Figure 7b, the curves for the increasing amount of ink, given in droplets (1 droplet ≈ 8 μL), are shown. The strongest variation of the shrinkage characteristics is for the smallest amount of ink. This correlates with the scenario of humidity penetration from the ambient air into the test ink. The larger the amount of the ink, the lower the water concentration and, consequently, the smaller the SFE change gauged by the test ink. This scenario is also in agreement with the observation that the difference between curves for one and two droplets is much larger than between the curves for two and three droplets.

The differences for the 10 s point are in the range of 10%, which is significant for the AIR method. To minimize the error resulting from this effect, the amount of ink should be proportional to the activation area, to keep the thickness of the ink film constant. The pragmatic rule for the experiments in this study is that the activation areas below 300 mm${}^{2}$ are visualized with a single ink droplet, and, for activation areas larger than 300 mm${}^{2}$, with two droplets.

#### 3.1.4. Influence of Ink SFE Value on Shrinkage Characteristics

Figure 8 shows the relative (all values of one curve are divided by its maximum value) shrinkage characteristics for the test inks in the range from 46 to 72 mN/m. They correspond to the activation area points from the plot in Figure 6b. The variation of the shrinkage characteristics from 46 to 58 mN/m, shown in Figure 8a, increases with increasing gauged SFE value, which correlates with an increase in the volume percentage of the formamide in the 2-ethoxyethylene/formamide test ink. In this case, the hygroscopic properties of the the formamide can explain this tendency. The higher volume percentage of the formamide in the test ink, the faster the absorption of the water from humid air in hygroscopic formamide and, consequently, the stronger the change of the gauged SFE. However, a geometrical effect can also play an important role. With the increasing total area of the ink patch, accompanying the test ink SFE value reduction, the same linear shift of the patch boundary during shrinking results in less relative area change.

Figure 8b shows the shrinkage characteristics recorded for the mixture of formamide with water. A shrinkage mechanism different to that described earlier must be considered for an explanation of these curves, because the presence of water in the mixtures excludes a strong change due to water absorption. The faster evaporation of formamide than water could be speculated, but the shrinkage characteristics for test ink gauged with 72 mN/m (100% water) require different explanations most probably based on an interaction between water and the activated HDPE surface.

#### 3.1.5. Renewed Ink Application

For a better understanding of the mechanisms of the ink patch shrinkage, the shrinkage characteristics were collected after renewed test ink application on the same patch. This experiment should show whether the origin of the shrinkage is related to the change of substrate surface properties or ink properties. In Figure 9a, the shrinkage characteristics for the fresh ink patch and the same ink patch refreshed two, four, and six times by the new test ink are shown. It can be observed that the starting points of all four shrinkage characteristics are quite close to each other. This means that the application of fresh ink causes gouging of the starting value of the visualized surface, which allows the conclusion that no substantial change in the activation area between taking two subsequent characteristics occurs. The conclusion of this experiment is that the changes in the test ink properties themselves are responsible for the shrinkage of the test ink patch.

To ensure that the shrinkage characteristics start with test ink without water admixture, the series of shrinkage characteristics were collected for fresh ink applied on the activated area after the removal of the previously applied ink. The test ink was removed by dabbing off with cellulose tissue. In Figure 9b, three such characteristics are shown for inking after one, three, and five ink removal/application cycles. It can be observed that the starting point of all characteristics is roughly at the same level, confirming the thesis that no change of the 58 mN/m activation area occurs. The observable change is the shift of the asymptotes to which the characteristics converge. They decrease with the number of whippings and re-applications of the test ink. The possible explanation of such behavior is the decrease of the maximum SFE on the HDPE surface after each wiping of the test ink. It is known that PCT (physical contact treatment), for example by the use of cellulose tissues, causes strong hydrophobic recovery. It was documented for PS, COP and PDMS in [61] that mechanical rubbing the activated surfaces causes the hydrophobic recovery. The ink removal with tissue from the HDPE surface can partially affect the layer of activation and cause the reduction of the SFE values. The flatter surface distribution of SFE results in a larger area decrease by a small change of the SFE gauged by the test ink.

#### 3.1.6. Hydrophobic Recovery

Typically, the hydrophobic recovery refers to the decrease of the SFE during long-term storage. For example, on oxygen-plasma treated polyethylene surfaces, a strong hydrophobic recovery is described [62]. The surface free energy of 46 mJ/m${}^{-2}$ reached on LDPE after corona treatment decreases after 22 days of exposition to air, down to the SFE of 36 mJ/m${}^{-2}$ [63], compared with the SFE for non-treated LDPE of 31 mJ/m${}^{-2}$.

In this study, the hydrophobic recovery refers to the reduction of the visualized activation area as a function of storage time. Figure 10 shows such a dependence for HDPE surface treated in configuration D. No hydrophobic recovery is stated in the time scale of the AIR measurement.

Despite the hydrophobic recovery documented for HDPE, no significant reduction of the activation area can be observed after storage of the activated substrates over an extended period. The results in Figure 10 document the changes over 300 h. The trend line shows in this time a decrease of 3 % of the activation area per 100 h. This result is not contradictory to the previously reported hydrophobic recovery results because those are related mainly to the decrease of the maximum value of the SFE reached after plasma treatment. Since the SFE of 58 mN/m used as a threshold for visualization in this work, which is much less than the maximum value of SFE obtained after treatment in configuration D, reaching the SFE of up to 72 mN/m (see Figure 6), the contour with 58 mN/m does not have to be affected much by the decline of the maximum SFE.

### 3.2. Characterization of Plasma Sources

To demonstrate the plausibility of the results collected with the AIR method, typical examples of operational characteristics of different APPS are presented and discussed. These are the dependence of the activation area (58 mN/m test ink on HDPE) on the treatment time and the distance from the plasma source.

#### 3.2.1. Dependence on Treatment Time

Fricke et al. [64] showed that the width, defined by the use of the profiles of the contact angle, of the activation area produced on the surface of polyethylene with APPJ non-linearly increases with treatment time and shows a tendency to saturate for very long treatment times. A similar behavior is observed for activation areas generated for both the PDD type plasma jets (see Figure 11a) and the PAA based plasma jets (see Figure 11b).

The two curves in Figure 11a show the dependences of the activation area on the treatment time for configurations B and C, respectively. The main difference between the operating conditions of these two configurations is different airflow. Configuration B is equipped with a stronger fan. The reduced dilution of the chemically active species in configuration C results in a higher activation efficiency, which is documented by a 17% increase in the activation area.

Configuration A is operated with power two orders of magnitude higher than that of B and C. The consequence is a much higher plasma temperature, which ranges from a few hundred °C in the diffuse plasma zone, to several thousand °C in the arc zone. This thermal difference causes a major difference in plasma chemistry. While the PDD produces ozone and active oxygen as the main chemical species, almost no ozone and high concentrations of nitrogen oxides, such as nitric oxide (NO) and nitrogen dioxide (NO${}_{2}$), are measured in gaseous products of the PAA plasma jet. Despite this difference, the PAA based plasma tools very efficiently activate the HDPE surface, with maximum SFE reaching 68 mN/m, corresponding to the contact angle of DI water of 35°. With such maximum value of SFE, the formamide gauged for 58 mN/m can be used for the quantitative evaluation of such a plasma jet.

The treatment area for configuration A was determined as a function of the treatment time and is visualized in Figure 11b. Even for the shortest pulse of 50 ms, an activation area of 63 mm${}^{2}$ is determined. For plasma switched on longer than 50 ms, not only the activation area can be visualized by test ink, but also a glossy HDPE surface becomes visibly mat. Such surface changes can be explained by a preferential material abrasion of low molecular weight materials, which produces changes in surface topography [65]. An interesting point on the time scale is the cross-over of the mat area curve and the activation area curve, occurring at 470 ms. For treatment times longer than 0.5 s a partial melting in the center of the activation area can be observed. The picture of the molten zone in the middle of the mat zone on the HDPE substrate is shown in Figure 12. Such molten areas have much lower SFE compared with a non-molten surface. For treatment times longer than 2 s, the molten zone becomes not wettable with formamide 20–30 s after the ink application. Such a case of partial wetting is shown in Figure 3, picture A${}_{2}$. After the reference time of 10 s, the molten zones were wettable. Consequently, the points for 3 and 5 s are included in the plot.

#### 3.2.2. Influence of the Substrate Distance

A further physical factor strongly affecting the activation area is the distance between the plasma source and the substrate. For plasma sources in configurations B and D, the activation area decreases with the distance increasing over 5 mm (see Figure 13a). This can be explained by increasing the dilution of the chemically active species and weaker electric fields enhancing the activation. The activation area for configuration B is roughly twice as high as that for configuration D because its operating power is higher by a factor of two. It can be also observed that the treatment with configuration B is possible at a larger distance of about 15 mm than for D, decaying at about 12 mm.

Much longer activation zones show the plasmas generated in configuration E operated in ambient air and F operated in argon flow. The activation area of both configurations is represented by red and blue lines in Figure 13b, respectively. The points of decay (zero points of the fitting lines) are 21 and 26 mm, respectively, compared with 15 mm for configuration B. Configuration E is operated without airflow. Consequently, the reason for the elongation of the plasma zone can be the more focussed electric field produced by the needle electrode. The even longer plasma jet in configuration F is caused by the argon flow. It contains long-living activated species, which can be transferred over a longer distance. It also allows us to sustain plasma at much lower electric fields, due to its lower break-down voltage. These tendencies are in agreement with well known results for other types of cold APPJs [66] and confirm the plausibility of the AIR results.

## 4. Conclusions and Outlook

A novel method for the evaluation of the activation area produced on polymer surfaces by atmospheric pressure plasma jets is proposed. The activation image recording (AIR) with a digital camera is used for the collection of shrinkage characteristics of activation zones wetted by the test ink.

This study demonstrates that AIR can be used as a diagnostic technique for the performance evaluation of atmospheric pressure discharges at different working conditions. It is also shown that it is suitable for the comparison of strongly different types of APPJs.

HDPE is selected as the best suitable material for test substrates thanks to: (i) its wide range of SFE achieved after APPJ treatment; (ii) low hydrophobic recovery; (iii) availability as a polymer without additives; (iv) moderate cost; and (v) high popularity as a reference material for plasma studies.

For activation area visualization, the test ink gauged for 58 mN/m (pure formamide liquid) is selected because: (i) the influence of changes of the proportion of two liquids with different volatility can be avoided; (ii) the SFE value is almost in the middle between the SFE of non-treated HDPE (35 mN/m) and the maximum achievable SFE of (72 mN/m); and (iii) formamide is defined as a component of the test inks in several important international standards.

The reference time on shrinkage characteristics is 10 s after distribution of the test ink on the HDPE surface. It is a compromise between a large absolute error in the area determination for a very short reference time and the influence of material and environmental factors and increasing relative error for a very long reference time.

The optimal treatment time should be selected depending on the kind of discharge and scales inversely proportional with plasma power. For example, to achieve a good resolution of the AIR results for the 700 W plasma device, the plasma treatment time in the ms range is needed. For PDD and other CeraPlas™ driven discharges, the treatment time of 10 or 20 s is optimal.

It is shown that the hydrophobic recovery, defined as change of the activation area with storage time after HDPE treatment with CeraPlas™ based device, is very slow: 3% per 100 h.

The origin of the short-term changes is in the temporal variations of the test ink properties. The most probable reasons for such variations are: (i) the change of stoichiometry of the two-component test ink due to different evaporation speeds of the components; (ii) the absorption of water from air humidity in the pure formamide test ink.

To achieve exact, statistically sound and reproducible results with the AIR method, some assumptions and rules of handling must be fulfilled.

The AIR results are valid only for intensive discharges, when 72 mN/m saturation on a large part of the activation area on HDPE is reached;the treatment time should not exceed the limit for the thermal damage of the HDPE surface;only a fresh test ink should be used, and the test ink vial should be opened only for a short time of ink application;the amount of the test ink should be adjusted to the size of the activation area; andthe AIR measurements should be conducted at room temperature and medium humidity.

Even though the physical and chemical mechanisms of the test ink patch shrinkage are not explained in detail, the shrinkage characteristics are successfully used for solving engineering problems during the development and evaluation of the novel plasma tools. Using AIR, the authors approached the following engineering tasks:determination of the optimum operating conditions for the maximum surface activation speed;investigation of the influence of the constructional changes on the APPJ performance;determination of equivalent working point for plasma tool replacing a different one;the investigation of the performance changes of the APPJ in course of an endurance test; andthe analysis of the influence of the type of discharge on the hydrophobic recovery.

The results of this study show that further work on this subject is needed. Among others, the physical–chemical mechanisms of the time-dependent shrinkage of the test ink patches should be investigated in more detail. The experimental development could also further improve the accuracy of the AIR technique. One example is the automation of the test ink application, allowing us to dose an exact amount of the liquid and to determine exactly the starting point for the ink patch shrinkage process.

## Figures and Tables

**Figure 1 polymers-13-02711-f001:**
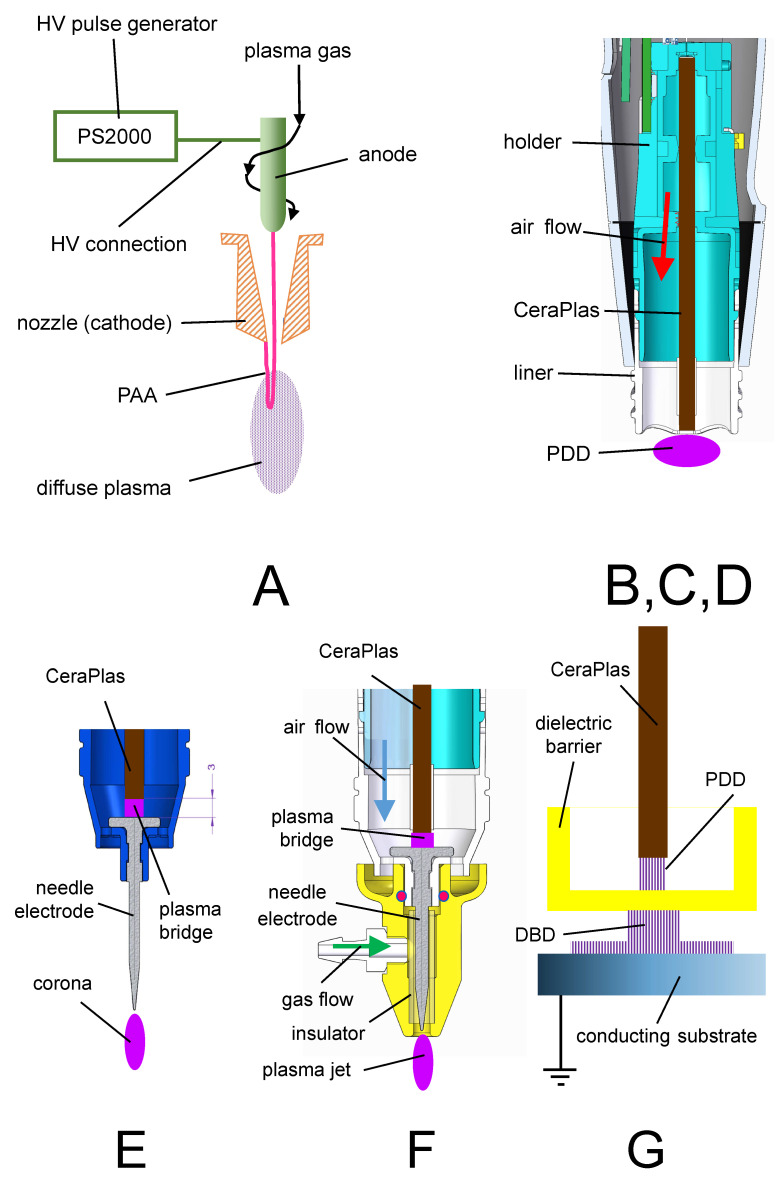
Discharge configurations. A—pulsed atmospheric arc (PAA) based APPJ, (B, C, D—piezoelectric direct discharge (PDD), E—piezoelectrically driven needle corona, F—piezoelectrically driven plasma needle discharge, and G—piezoelectrically driven dielectric barrier discharge (DBD).

**Figure 2 polymers-13-02711-f002:**
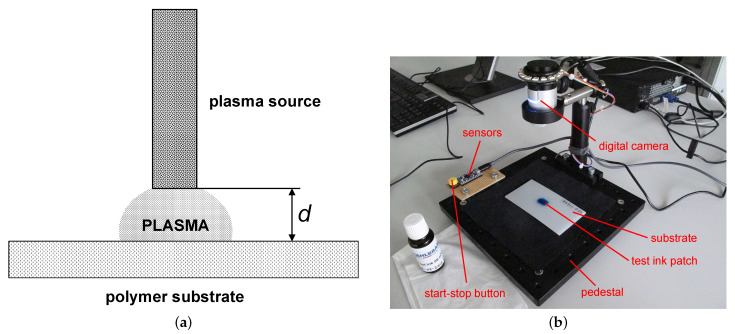
Experimental setup. (**a**) Generic setup for activation of the polymer substrate surface. The meaning of *d* depends on configuration A–G as defined in Table 1. (**b**) Picture of setup for AIR.

**Figure 3 polymers-13-02711-f003:**
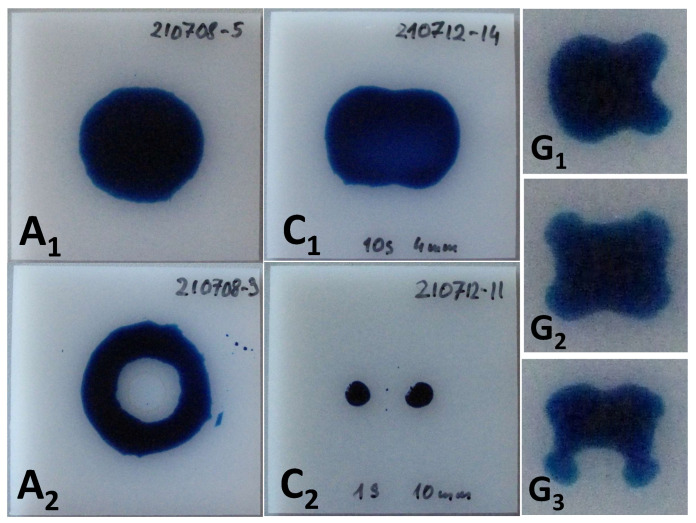
The shapes of the activation area visualized by the 58 mN/m test ink on the HDPE substrate for configurations A, C and G. The sizes of the substrates seen in pictures for A and C are 50 mm × 50 mm. The treatment times are 100 ms, 3 s, 10 s, 1 s and 20 s for A${}_{1}$, A${}_{2}$, C${}_{1}$, C${}_{2}$, and G${}_{1-3}$ respectively.

**Figure 4 polymers-13-02711-f004:**
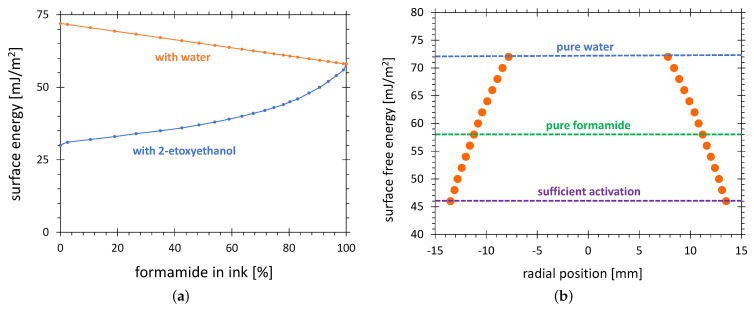
Determination of the SFE by use of the test inks. (**a**) The SFE gauged by the test ink consisting of formamide mixed with 2-ethoxyethanol (blue curve, according to [49]) and formamide mixed with water (red line, linear approximation) as a function of formamide volume percentage; (**b**) The typical radial distribution of the SFE after HDPE treatment in configuration C. The SFE levels for water and formamide test inks, and for the levels sufficient for printing on HDEP, are depicted with dashed lines.

**Figure 5 polymers-13-02711-f005:**
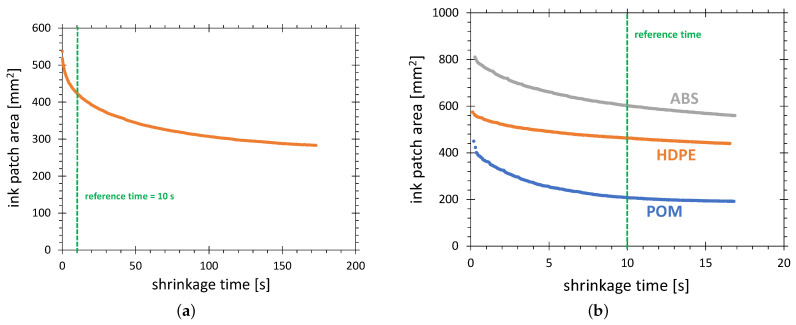
Shrinkage characteristics of the 58 mN/m test ink patch (**a**) taken for 3 min on HDPE, and (**b**) taken 17 s on three substrate materials: ABS, HDPE and POM. The discharge configuration C was used.

**Figure 6 polymers-13-02711-f006:**
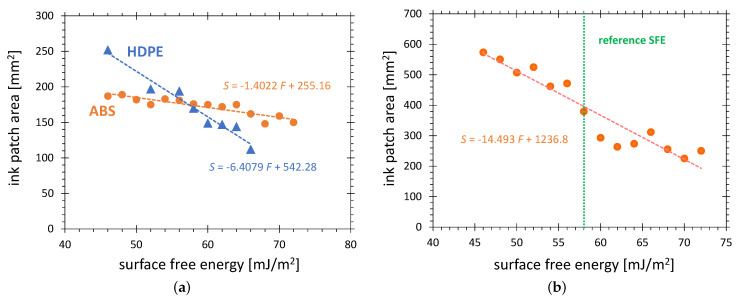
The visualized activation area on a substrate surface as a function of SFE calibration of the test ink. (**a**) Discharge configuration G was used with the treatment time of 20 s on HDPE and 10 s on ABS; (**b**) Discharge configuration C was used for 10 s treatment on HDPE.

**Figure 7 polymers-13-02711-f007:**
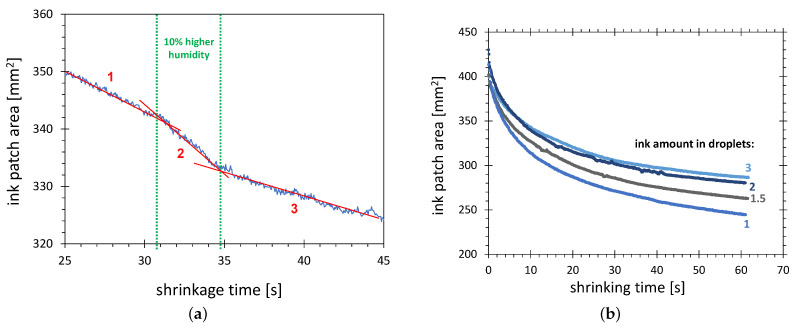
Shrinkage characteristics of the 58 mN/m test ink patch on HDPE plasma-treated in configuration C. (**a**) The segment showing the influence of increased air humidity; (**b**) The influence of the different amounts of applied ink.

**Figure 8 polymers-13-02711-f008:**
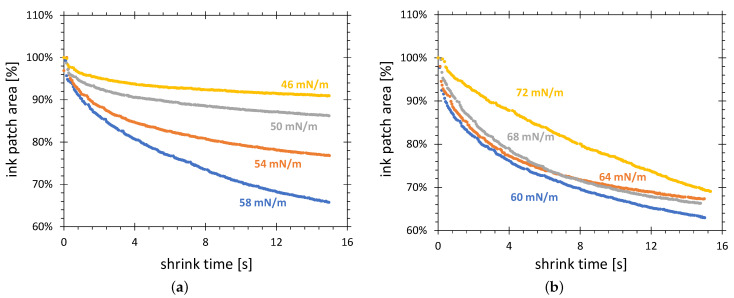
Comparison of relative shrinkage curves for test inks produced as mixtures of formamide with (**a**) 2-ethoxyethylene, and (**b**) water. The HDPE substrate was treated in configuration C.

**Figure 9 polymers-13-02711-f009:**
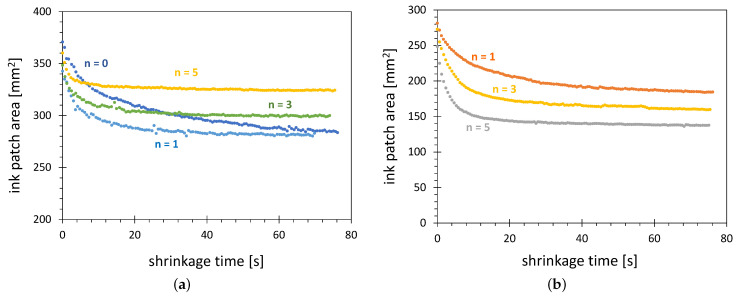
Shrinkage characteristics of the 58 mN/m test ink patches (**a**) directly after first ink application, after 1, 3 and 5 ink re-applications and (**b**) after 1, 3 and 5 ink wiping and renewed application. Treatment conducted in configuration D with the power of 4.5 W at the distance of 5 mm.

**Figure 10 polymers-13-02711-f010:**
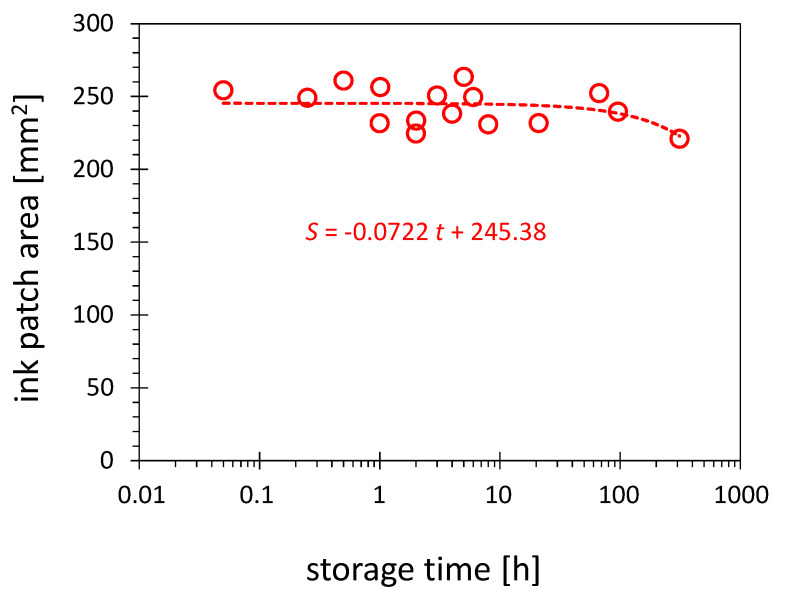
Influence of the storage time on the activation area. The plasma treatment of HDPE substrates was conducted in configuration D and visualized with the 58 mN/m test ink.

**Figure 11 polymers-13-02711-f011:**
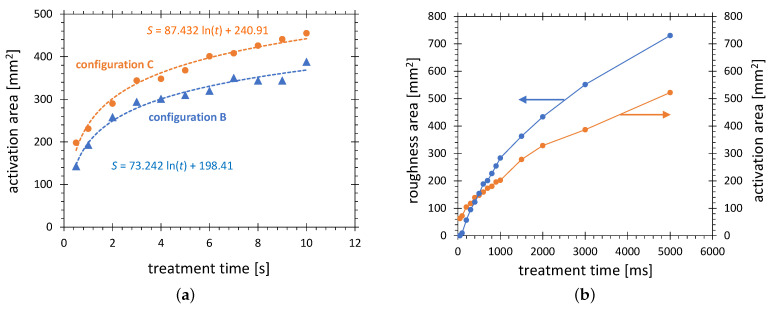
Dependence of activation area (58 mN/m, HDPE) on the treatment time. The plasma treatment is performed in configurations (**a**) B and C, and (**b**) A, operated at conditions from Table 1, respectively. The area of thermally induced loss of glossiness is included for comparison.

**Figure 12 polymers-13-02711-f012:**
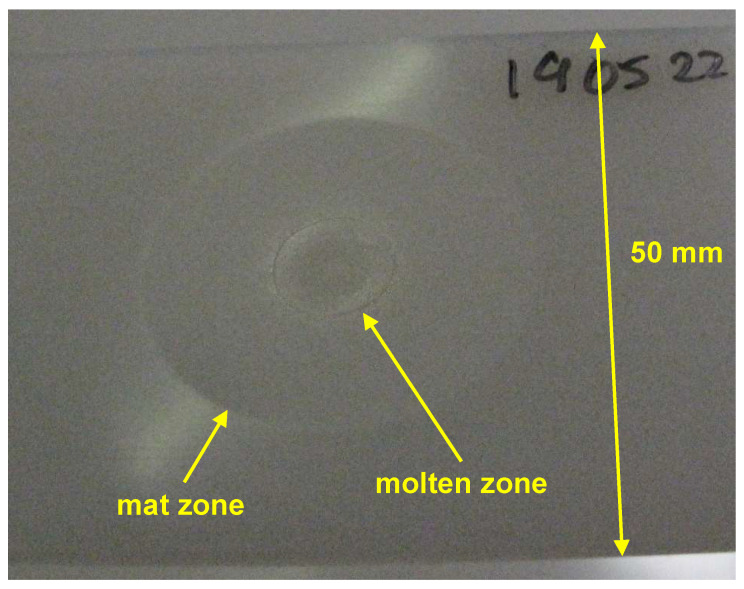
The picture of the HDPE substrate taken after treatment in configuration A operated at conditions from Table 1. The treatment time is 3 s.

**Figure 13 polymers-13-02711-f013:**
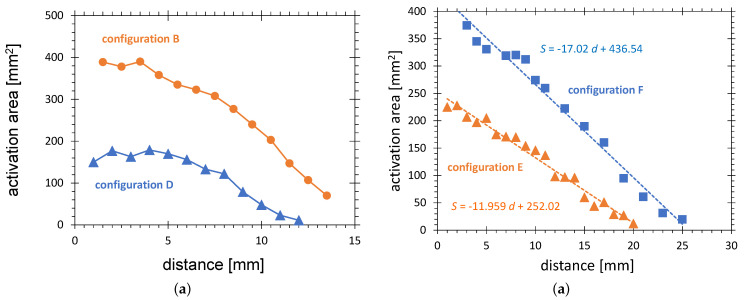
Dependence of the activation area (58 mN/m, HDPE) on distance for (**a**) configuration B and D, and (**b**) configuration E and F.

**Table 1 polymers-13-02711-t001:** Discharge configurations and default operating parameters. Distance means the distance between the substrate and the orifice of the nozzle for A, the tip of the CeraPlas™ for B and C, the tip of the needle electrode for E, F, and the outer surface of the dielectric barrier for G.

Case	Device	Nozzle	Power [W]	Frequency [kHz]	Distance [mm]	Gas	Gas Flow [SLM]
A	plasmabrush^®^ PB3	A450	700	54	25	CDA	35–80
B	piezobrush^®^ PZ2	CeraPlas™F	8.0	50	6	air	∼20
C	piezobrush^®^ PZ3	CeraPlas™F	8.0	50	6	air	∼10
D	CeraPlas™package	CeraPlas™HF	4.5	82	5	CDA	
E	piezobrush^®^ PZ3	needle corona	8.0	50	3–20	air	
F	piezobrush^®^ PZ3	plasma needle	8.0	50	3–25	Ar	3
G	piezobrush^®^ PZ3	DBD	8.0	50	1.0	air	

## Data Availability

The data can be obtained on request from the first author.

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
