# Peer review of "Visualization of Activated Area on Polymers for Evaluation of Atmospheric Pressure Plasma Jets"

_polymers, 2021, doi:10.3390/polym13162711_

Round 1

Reviewer 1 Report

Comments and Suggestions for Authors

This manuscript was dealt with visualization of activated area for evaluation of atmospheric pressure plasma jets of activation image recording (AIR) technique. The manuscript was not written well-organized and the results were very interesting. This manuscript was suitable to publish in the ‘Polymers’. However, there are so many grammatical errors revisions/corrections which the authors should consider prior to the publication. This manuscript should be published after major revision.

1.English should be improved for the better understanding. Many sentences contain formal and grammatical errors. Although I appreciate that English might not be the first language of the authors, it is necessary to improve the readability of the text a bit. The manuscript should be totally revised by the native speaker or other researchers.

For examples,

Page 1, author: ~ Dariusz Korzec , Thomas~→add to symbol (*) for correspondence ~ Dariusz Korzec* , Thomas~

Page 1, affiliation: ~ relyon plasma GmbH,~ → ~ Relyon plasma GmbH,~

Page 1, Line 8: ~ The time dependent area (shrinkage characteristic)~ → formal and grammatical errors

Page 2, Line 9: ~ for four different types of plasma: (i)~ → ~ for four different types of plasma (i)~

delete “:”

Page 2, Line 10: ~ pulsed atmospheric pressure APPJ with power~ → ~ pulsed APPJ with a power~

Page 2, Line 12: ~are shown, what kind of characterization~ → ~are shown what kind of characterization~ delete “comma (,)”

Page 2, Line 15-16: Keywords: atmospheric plasma; dielectric barrier discharge; piezoelectric direct discharge; surface free energy; test ink; surface activation; à Keywords: atmospheric plasma, dielectric barrier discharge, piezoelectric direct discharge, surface free energy, test ink, surface activation → delete “;”

Page 2, Line 22: ~ Activation Image Recording (AIR)~ → ~ Activation image recording (AIR)~

Page 3, Line 47-48: For high power density applications, such concepts are developed like inductively coupled plasma ICP ~ → ~inductively coupled plasma (ICP) ~

~and capacitively coupled plasma CCP ~ →~and capacitively coupled plasma (CCP) ~

Page 3, Line 40: ~ The Non-Equilibrium Atmospheric Pressure Plasmas (APP)~ → ~The non-equilibrium atmospheric pressure plasmas (APP)~

Page 3, Line 52: ~ inµm scale. Recently~ →~ in µm scale. Recently,~

Page 3, Line 73: ~ In this study the activation area is~ → ~ In this study, the activation area is~

Page 3, Line 75: ~in detail in [35]. ~ → ~in detail [35]. ~

Page 6, Line 180: ~ different types of atmospheric pressure plasma sources.~ → ~ different types of APP sources.~

Page 6, Line 182: 3.1.1. Shrinking of the test ink patch → 3.1.1. Shrinkage of the test ink patch

Page 8, Line 244: ~very week plasma~→ ~very weak plasma~

  1. Especially, specific words were too often used in this manuscript (For example, ‘piezobrush’, ‘by use of’ and etc.). I recommend that the authors should use abbreviation appropriately (For example, ~ the commercial devices piezobrush PZ2 and PZ3 of Relyon Plasma~)

          In addition, common errors were too often used in this manuscript (For example, ‘FSE’ on page 5, line 137 and etc.).

          I recommend that the authors totally should correct in manuscript.)

  1. All references should be totally corrected according to polymers format.

For examples,

Kogelschatz, U.; Baessler, P. Determination of nitrous oxide and dinitrogen pentoxide concentrations in the output of air-fed ozone generators of high power density. Ozone: Science & Engineering 1987, 9, 195-206.

Chapter 2.1 should be improved for the better understanding. I recommend that the authors combine the Table I for all experimental conditions as follows attached PDF file. For examples, (Table I)

  1. In Figure 1, you should improve for the better understanding and add schematic diagram and photo-image in revised manuscript for experimental set-up as follows attached PDF file. For examples, (Figure 1b)
  2. What means a stead illumination in manuscripts on page 5, line 122 ?
  3. What means a “The ABS, POM and HDPE are considered as suitable candidates as a substrate material for AIR method.” in manuscripts on page 8, line 231-232? If necessary, you should add to possible explanation and related reference paper. 

Author Response

Dear Reviewer,

thank you very much for the very constructive review.

I have followed all your corrections.

Best regards

D. Korzec

Reviewer 2 Report

The paper reports plasma-stimulated activation of polyethylene. It looks like an advertisement for commercial plasma devices.  The title is misleading: not only HDPE was used as testing material. The authors refer to numerous previously published papers, but the review is outdated. In fact, only three papers published since  2020 were cited, all authored by the same group. This is a hot topic of current research, so the authors should refresh the literature survey – they may be able to find detailed papers on the surface activation of polymers with atmospheric plasma. The experimental part is poor – The authors represent numerous diagrams but do not report on the specific experimental conditions. They mention powers, but not the power per unit area, the average power or the distance between the plasma source and the substrate. Much discussion is on the limitations of the methodology (AIR method), which is not very useful for the scientific community. The increased surface activation area with increasing treatment time is easily explained by gradients in the reactive plasma species likely to occur in atmospheric-pressure plasmas. The authors should define the “roughness area” and support the definition by AFM or SEM images. Much discussion is on the ageing of test liquids – the effect known to the scientific community. I miss systematic measurements and scientific explanation of the observed results. To summarize: the manuscript does not meet the standards of a scientific paper.

The minor comments include:

  1. The term “plasma” in line 9, page 1 should be replaced with “discharge”.
  2. I don’t understand the meaning of the statement, “Using this value of surface energy allows also for a large dynamics of the method.” in the abstract.
  3. Also in the abstract, the powers for two types of discharges are stated and not mentioned for another two types. What sort of a jet operates at 700 W? Any plastics exposed to such a powerful device would melt immediately.
  4. What is “plasma nozzle in Figure 1a? Discharge tube, powered electrode or something else?
  5. There should be scale bars in Figure 1b.
  6. Figure 2 should be moved after 2.4.1.
  7. Most citations not dealing with plasma treatment of polymers should be omitted.
  8. There are spelling or grammatical mistakes throughout the text.
  9. Avoid citing conference proceedings and similar sources – try to cite only peer-reviewed scientific papers.

Author Response

Dear Reviewer,

thank you for your constructive criticism. I have done my best

to fulfill your requirements.

Best regards

D. Korzec

Reviewer 3 Report

This is a novel and very interesting investigation of the applicability of a new method developed by authors for characterization and testing of various atmospheric-pressure plasma devices by monitoring the size of the plasma-activated area on the reference material. The activated area is defined by wetting it with a special ink. The authors made a detailed investigation on the selection of the appropriate reference material giving the optimal response when applying ink. Furthermore, the authors also investigated the sensitivity of a liquid to ageing because of environmental effects (humidity…) as well as ageing of the plasma-treated substrate. The manuscript is well written, and it can be published after minor modifications.

  1. Introduction, line 50: What is “clean” plasma?
  2. Introduction, line 55: “comment” or “common”?
  3. Materials and Methods: The authors used four “strongly” different types of plasma. Can you shortly describe what are the major differences or maybe give a scheme of all four plasma configurations?
  4. The authors used four different devices with different tips. Can you use various tips on the same device? Do these tips have different nozzle configurations causing different shapes of the activated area, as shown in Figure 1b? Why do you have five images in Figure 1b and not four? What is the size of the images in Figure 1b?
  5. Why did you chose formamide ink if it is so sensitive to water uptake and other environmental issues? Is it because it is having just the right SFE?
  6. When using APPJ for polymer treatment, we can observe different sizes (even 2 cm) of an activated area with a gradient of wettability being maximum in the middle and decreasing toward the outer edges of the treated area. In practical applications, when using your method for comparing different configurations of plasma jets for polymer activation, each jet can cause not only a different size/shape of the activated area but it may also produce different wettability in the middle of the treated spot. How does this affect your method if we want to use it for the comparison of different plasma devices?

Author Response

Dear Reviewer,

thank you for your constructive criticism.

I responded positively to all your requirements.

Best regards

D. Korzec

Round 2

Reviewer 1 Report

According to the reviewers’ comments, the manuscript are totally revised and the revised paper are well-organized. Therefore, I think this article can be published in a present form at the ‘Polymers’.

Reviewer 2 Report

The authors did their best to improve the manuscript. Still, my opinion is that the paper lacks a scientific component.